# More than Meets One Core: An Energy-Aware Cost Optimization in Dynamic Multi-Core Processor Server Consolidation for Cloud Data Center

**Huixi Li** [1,2]**, Langyi Wen** [1]**, Yinghui Liu** [3] **and Yongluo Shen** [1,2,*]

[1]  School of Information Science, Guangdong University of Finance and Economics, Guangzhou 510320, China
[2]  Guangdong Intelligent Business Engineering Technology Research Center, Guangdong University of Finance and Economics, Guangzhou 510320, China
[3]  School of Chinese Language and Literature, Nanjing Xiaozhuang University, Nanjing 211171, China
[*]  Correspondence: sylkyo@gdufe.edu.cn

**Abstract:** The massive number of users has brought severe challenges in managing cloud data centers (CDCs) composed of multi-core processor that host cloud service providers. Guaranteeing the quality of service (QoS) of multiple users as well as reducing the operating costs of CDCs are major problems that need to be solved. To solve these problems, this paper establishes a cost model based on multi-core hosts in CDCs, which comprehensively consider the hosts' energy costs, virtual machine (VM) migration costs, and service level agreement violation (SLAV) penalty costs. To optimize the goal, we design the following solution. We employ a DAE-based filter to preprocess the VM historical workload and use an SRU-based method to predict the computing resource usage of the VMs in future periods. Based on the predicted results, we trigger VM migrations before the hosts move into the overloaded state to reduce the occurrence of SLAV. A multi-core-aware heuristic algorithm is proposed to solve the placement problem. Simulations driven by the VM real workload dataset validate the effectiveness of our proposed method. Compared with the existing baseline methods, our proposed method reduces the total operating cost by 20.9~34.4%.

**Keywords:** cloud computing; multi-core processor; server consolidation; VM migration; SLAV; energy consumption

## 1. Introduction

The world is entering a post-coronavirus era. Since countries and multinational cooperative organizations still have not formed a unified, reliable, and effective means of epidemic prevention, a local epidemic that could break out at any time brings a high risk of spreading to the world. This situation has forced people to further embrace cloud computing, migrating much of their economic, social, and personal activities online. For example, about 82% of Hong Kong businesses plan to maintain remote work in the post-COVID-19 era [1]. This trend has brought opportunities for cloud computing, as well as management pressure. According to estimates, the current compound annual growth rate of the Hong Kong data center market value is 12.6%, which means that the value will reach HKD 4.12 billion by 2026 [2]. The increase in market value means that practitioners need more cost investment.

Increasing the resource rate of cloud data centers (CDCs) is one of the most effective means to reduce management costs, but there is a conflict between reducing costs and the performance that cloud service customers receive. To improve resource usage, virtual machines (VMs) or containers assigned to users must be highly concentrated on physical hosts. However, a high degree of centralization brings a high degree of resource competition. When the competition is too intense, the host may be overloaded, thereby reducing the performance and user experience of VMs. To ensure the user experience, service level

agreements (SLAs) are used to quantitatively describe the corresponding quality of service (QoS). If the SLA cannot be maintained, the QoS is threatened, a and SLA violation (SLAV) is generated. When a SLAV appears, cloud service providers (CSPs) need to provide compensation to users as punishment for failing to meet user performance requirements. Currently, server consolidation is used to dynamically adjust the load balance between hosts in a CDC. Server consolidation periodically checks the load of hosts in the cluster and initiates VM migration to achieve load balancing, thereby maintaining a balance between resource utilization and performance.

Multiple works designing server consolidation solutions assume that the physical host is equipped with a single-core CPU, and multi-core processors have long been popular in personal entertainment, scientific research, and data centers. A CPU package consists of multiple dies, and each die encapsulates multiple cores. Due to the involvement of inter-core communication, inter-die communication, and other CPU components, the power consumption of a multi-core CPU is much higher than that calculated by the single-core CPU power consumption model. Therefore, the server consolidation model based on a single-core processor cannot accurately describe the user's energy demand. In addition, CSPs need to provide additional overhead to maintain VM migration in server consolidation and possible SLAV compensation. In this paper, we establish a server consolidation cost model based on the use of multi-core processor memory resources, VM migration, and SLAV compensation and propose corresponding solutions to achieve a balance between cost and performance. Our contributions are as follows:

(A)  We formally define a host power consumption model based on multi-core CPU and memory resource usage and describe the cost of VM migration and SLAV on this basis. After proposing the cost model, we give the corresponding optimization problem.

(B)  A denoise autoencoder-based filter is used to denoise the VM workload trace. Subsequently, we use the SRU-based RNN method to predict the workload of VMs. Based on the predicted results, a host load detection strategy is proposed that considers both current and future load conditions.

(C)  To minimize the total cost of server consolidation, we propose a VM selection strategy and a VM placement algorithm. These methods take into account the scheduling and placement of VMs between different cores of the same CPU and between different CPUs of different hosts, as well as the current and future requirements of VMs for different resources.

(D)  We conduct simulations to evaluate the performance of our proposed solution MMCC. The simulations' results indicate that MMCC can reduce host energy consumption by 10~43.9%, SLAV cost by 33.5~51.7%, and total cost by 20.9~34.4% compared to the baseline methods.

The remainder of the paper is organized as follows. In Section 2, we survey the related work. In Section 3, we formalize the cost model and define the corresponding optimization problem. In Section 4, we propose a heuristic algorithm to solve this problem. In Section 5, we evaluate the performance of our proposed method using trace-driven simulations based on real VM workloads. In Section 6, we include the paper and discuss future works.

## 2. Related Work

In this section, we survey the CDC cost model related to server consolidation and the corresponding solutions.

### 2.1. Server Consolidation Cost Models

Based on single-core CPU usage or performance, a large number of works on server consolidation proposed host energy models [3–12]. Nagadevi et al. [13] proposed a VM placement algorithm based on multi-core processors, but they did not consider factors related to dynamic consolidation, energy consumption, and cost throughout the data center life cycle. The above work also did not consider the energy consumption of the processor at the die level and the chip level.

In addition, the composition of a host's energy consumption is not only related to the CPU factor. Therefore, several works have proposed multi-resource utilization-oriented host energy models [14–19]. However, these models only consider the energy consumption when the host acts as an independent object and do not consider the additional energy consumption of the VM migration due to the increase of the host load during server consolidation.

To ensure user performance and service quality, Buyya et al. [20] proposed a CPU-based SLAV calculation method, which was widely adopted in many subsequent works [21–29]. However, the quality of service (QoS) of users when using VMs cannot be measured only by CPU performance, and SLAV must involve the use of multiple resources.

### 2.2. Server Consolidation Solutions

Buyya et al. [20] first proposed the classic four-step server consolidation solution. The first step is host load detection, which picks out overloaded and underloaded hosts in the cluster. The second step is VM selection for overloaded hosts. In order to reduce the host load and the occurrences of SLAV, suitable VMs are selected and added into a VM migration list. The third step is VM placement, which selects the suitable destination hosts for all objects in the VM migration list. After VM placement, underloaded hosts are handled. By migrating all the VMs on the underloaded host to other suitable hosts as much as possible and shutting down or switching these underloaded hosts to an energy-saving state, the host energy cost of the CDC can be further reduced. At present, most of the specific execution strategies for solving server consolidation are heuristic. Based on multiple resource constraints, Li et al. [30] proposed a server consolidation method that not only reduces energy consumption but also ensures QoS, but this method only guarantees the QoS of the users in terms of CPU usage. YADAV et al. [25] mainly considered the network overhead and proposed an adaptive host overloaded detection method and VM selection algorithm. Sayadnavard et al. [31] proposed a server consolidation method based on multiple resource constraints, but the optimization goal is to minimize the number of hosts used by the VM placement, and it ignores other types of costs. Yuan et al. [32] used the culture multiple-ant-colony algorithm to solve the server consolidation problem without SLAV constraints.

None of the models proposed in the above works simultaneously consider the costs associated with multi-resource usage, multi-core processors, multi-resource SLAV, and VM migration.

## 3. Cost Model and Problem Description

In this section, we first formally describe the multi-core processor-based cost model in server consolidation of CDC and then formulate a problem description based on this.

### 3.1. Cost Model

In CDC, the cost related to server consolidation mainly involves hosts, VM migrations, and SLAV compensation.

Before giving a specific cost model, we first describe the time and objects of the entire system. There are $N$ heterogeneous hosts in the CDC, forming the host set $H = h_1, h_2, \cdots, h_N$. The total amount of resources that a host $h_i$ can provide is marked as a scalar $C_i = (c_i^{cpu}, c_i^{mem})$, where $c_i^{cpu}$ and $c_i^{mem}$ are the total amount of CPU and memory resources, respectively. The CPU is multi-core; hence we have $c_i^{cpu} = (c_i^{core_1}, c_i^{core_2}, \cdots, c_i^{core_{cn_i}})$, where $cn_i$ is the number of cores in the processor on $h_i$. Generally speaking, we make $c_i^{core_1} = c_i^{core_2} = \cdots = c_i^{core_{cn_i}}$, where $c_i^{core_{cn_i}}$ is the total amount of computing resources that each core can provide. There are $M$ VMs running on these hosts, forming a VM set $V = v_1, v_2, \cdots, v_M$. When a user makes a VM $v_j$ request, the submitted resource requirements are marked as scalar $D_i = (d_j^{cpu}, d_j^{mem})$, where $d_j^{cpu}$ and $d_j^{mem}$ are the total requirements of $v_j$ for CPU and memory, respectively. We assume that each VM is a single-core task; that is, only the computing resources of a single core can be used by a certain VM.

The life cycle of a CDC $[0, LT]$ is divided into $L$ small and equal-length consecutive time segments $t_1, t_2, \ldots, t_L$, and each time segment has a length of $T$. In a certain time segment $t_k$, if a host $h_i$ is in working state, $\lambda_{i,k} = 1$, otherwise $\lambda_{i,k} = 0$. At this time, the amount that the host can provide for each resource is $R_{i,k} = (r_{i,k}^{cpu}, r_{i,k}^{mem})$, where $r_{i,k}^{cpu} = (r_{i,k}^{core_1}, r_{i,k}^{core_2}, \cdots, r_{i,k}^{core_{cn_i}})$, where $r_{i,k}^{core_{cn_i}}$ is the amount of resources that the $cn_i$-th core can provide in $t_k$. In $t_k$, the amount of resources demanded by the VM $v_j$ is denoted as $S_{j,k} = (s_{j,k}^{cpu}, s_{j,k}^{mem})$.

We summarize the total cost of a CDC for a given lifetime by analyzing the performance of each computing device in each time slice. In general, in addition to the operating cost of the hosts, it is also necessary to consider the cost of VM migration during each server consolidation and the penalty caused by the occurrences of SLAV. We will discuss them separately in the following summaries.

Host Cost Model

Given a host $h_i$, its running cost $C_{h_i}$ is mainly related to the electricity charge $EP$ and its power $p_{h_i,t}$ at a given time $t$, namely:

$$C_{h_i} = EP \times \int_0^{TL} p_{h_i,t} dt. \tag{1}$$

It should be noted that if $h_i$ is powered off or in a power-saving state, its power consumption is negligible, so it will not incur any electricity-related costs. The analysis [33] of VM traces in the Alibaba CDC shows that the demand for CPU and memory resources of VMs far exceeds that of disk and network I/O. In this paper, we consider that the power of a host is related to the CPU, memory, and other basic components (motherboard, network card, disk, etc.). We also consider the power consumption of basic components to be a fixed value, so the power consumption of CPU and memory is discussed below.

- CPU power model

Buyya et al. [20] leveraged a single-core-based host power model in server consolidation; that is, the power of the CPU is related to its only core. Modern processors are multi-core architectures. Multiple cores are packaged on multiple CPU dies. The general architecture of a multi-core CPU is shown in Figure 1.

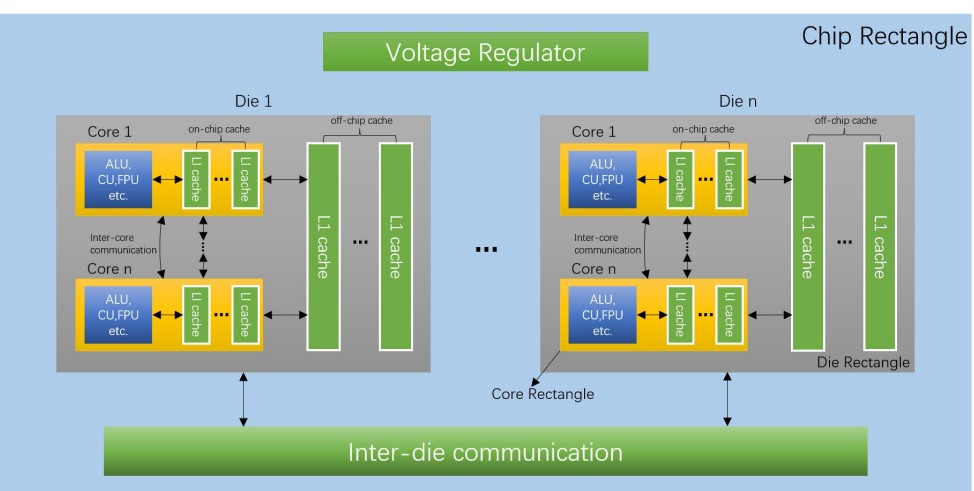

**Figure 1.** The general architecture of a multi-core CPU.

The total power consumption of the processor involves chip-level mandatory components, cores, die-level mandatory components, communication between cores, and communication between dies. In addition, modern processors employ energy-efficient mechanisms (such as Intel's SpeedStep) to optimize the power consumption of the CPU,

which means that the power consumption of the CPU is not linearly related to its usage. We describe the power description of a given CPU at a given moment as:

$$P_{cpu} = (1 - r) \times (P_{cm} + P_{dies} + P_{interdie}), \tag{2}$$

where $r$ is the energy-efficient factor, $P_{cm}$ is the power consumption of chip-level mandatory components, $P_{dies}$ is the power consumption of dies, and $P_{interdie}$ is the power consumption of inter-die communication. Next, we give the models of the above factors and energy consumption, respectively. In the case of not using an energy-efficient mechanism, the actual power when the $n$ cores of the processor perform calculations at the same time is $P_{act}$:

$$P_{act} = P_{cm} + P_{dies} + P_{interdie}. \tag{3}$$

In addition, we denote the total power of all cores as:

$$P_{cores} = \sum_{k=1}^{n} P_{core}^{k}, \tag{4}$$

where $P_{core}^{k}$ is the power consumption of the $k$-th core when other cores are idle and it is computing alone.

Basmadjian et al. [34] performed experiments to analyze the power consumption of chip-level mandatory components such as voltage regulators for:

$$P_{cm} = P_{cores} - P_{act} = s(v, f), \tag{5}$$

where $s$ is the capacitance function, $v$ is the voltage, and $f$ is the frequency.

$$s(v, f) = ce \times v^2 \times f, \tag{6}$$

where $ce$ is the effective capacitance [35].

Communication between dies occurs when cores on different dies access data at the same memory address. The power consumption of inter-die communication is:

$$P_{interdie} = \sum_{j=1|d_j \in D}^{|d|-1} ce \times v_j^2 \times f_j, \tag{7}$$

where $v_j$ and $f_j$ are the voltage and frequency of the corresponding cores on *diej*, $d_j$ is the set of active cores on the $j$-th die, and $D$ is the set of dies related to communication, and they are:

$$D = \{d_j | d_j \neq \varnothing\}, \tag{8}$$

$$d_j = \{core_{i,j} | u(core_{i,j}) > 0\}, \tag{9}$$

where $core_{i,j}$ is the $i$-th core on the $j$-th die, $u(core_{i,j})$ is the current utilization of $core_{i,j}$, $i \in [1, n_j]$, and $n_j$ is the total number of cores on the $j$-th die. We also have:

$$v_j = max\{v_{core_{i,j}} | u(core_{i,j}) > 0\}, \tag{10}$$

$$v_j = max\{f_{core_{i,j}} | u(core_{i,j}) > 0\}. \tag{11}$$

Equations (10) and (11) show that when there is only one active core on the $j$-th die, $v_j$ and $f_j$ of the $j$-th die are the voltage and frequency of the core.

The power of a single die can be described as:

$$P_{die}^{j} = P_{md}^{j} + P_{cores}^{j} + P_{off}^{j}, \tag{12}$$

where $P^j_{md}$ is the power consumption of die-level mandatory components, $P^j_{cores}$ is the power consumption of $n_j$ constituent cores, and $P^j_{off}$ is the power consumption of off-chip caches. We leverage Equation (5) to model $P^j_{md}$.

Inter-core communication occurs between multiple cores on a single die $j$. Therefore, the core-level power consumption model is:

$$P^j_{cores} = P^j_{dc} + P^j_{intercore},$$ (13)

where $P^j_{dc}$ is the power consumption of all active cores on $j$-th die, and $P^j_{intercore}$ is the inter-core communication power consumption between the active cores.

The power consumption of a single core $core_{i,j}$ is described as:

$$P^j_{core_{i,j}} = P^{core_{i,j}}_{exc} + P_{on},$$ (14)

where $P^{core_{i,j}}_{exc}$ and $P_{on}$ are the power consumptions of exclusive components (*e.g.*ALU) and on-chip caches of $core_{i,j}$, respectively. Based on the model in [20], we consider that $P^{core_{i,j}}_{exc}$ is linearly related to the utilization of the core, therefore:

$$P^{core_{i,j}}_{exc} = P^{core_{i,j}}_{max} \times \frac{u(core_{i,j})}{100},$$ (15)

where $P^{core_{i,j}}_{max}$ is the power consumption of $core_{i,j}$ at the maximum utilization, which can be calculated by the model in Equation (5).

The power consumption of on-chip caches is:

$$P_{on} = \sum_{i=1}^{s} P_{L_i},$$ (16)

where $s$ is the number of the on-chip caches, $P_{L_i}$ is the power consumption of on-chip cache $L_i$, which can be calculated by the model in Equation (5).

Hence, the power consumption of all active cores on the $j$-th die is:

$$P^j_{dc} = \sum_{i=1|core_{i,j} \in d_j}^{n_j} P_{core_{i,j}}.$$ (17)

By dynamically adjusting voltage and frequency and turning off temporarily unused components, the energy-efficient mechanism can effectively optimize processor power consumption. This part of the power consumption reduction is mainly affected by three factors: (1) components and communication between cores, (2) changes in the frequency of a single core, and (3) the number of cores. Here we define the three factors.

The first factor is

$$\alpha = 1 - \frac{P_{act}}{P_{cores}}.$$ (18)

The second factor is

$$\beta = \frac{\alpha}{f},$$ (19)

where $f$ is the given frequency. For a multi-core processor, we have:

$$f = average\{f_{core_{i,j}} | core_{i,j} \in d_{i,j}, j \in [1, m]\},$$ (20)

where $m$ is the number of dies.

The third factor is

$$\gamma = \begin{cases} \dfrac{\alpha}{k}, & k \geq 2, \\ 0, & otherwise, \end{cases} \tag{21}$$

where $k = \sum_{j=1}^{m} |d_j|$ is the total number of all active cores on the processor.

Based on Equations (18), (19), and (21), we obtain the power reduction factor:

$$r = \alpha + \beta + \gamma. \tag{22}$$

Based on the above analysis, the processor power consumption $P_{cpu}$ of a given host can be obtained. For the host $h_i$, the power consumption of its processor at a certain time $t$ is denoted as $P_{i,t}^{cpu}$. It can be said that $P_{i,t}^{cpu}$ is a function of the current utilization of each core on the processor.

- Memory power model

In all current public data traces of a VM, workload records in the CDC provide the memory usage of monitored objects within a certain period of time. Therefore, the current footprint $u_{i,t}^{mem}$ of the memory is used to estimate the power consumption $P_{i,t}^{mem}$ of the host $h_i$ at a given time $t$:

$$P_{i,t}^{mem} = P_{idle,i}^{mem} + \alpha_i^{mem} \times u_{i,t}^{mem}, \tag{23}$$

where $P_{idle,i}^{mem}$ is the memory power consumption when $h_i$ is idle, and $\alpha_i^{mem}$ is the memory power factor. According to the analysis by Esfandiarpoor et al. [27], when $\alpha_i^{mem} = 0.3W/GB$, the power consumption of a DDR memory system can be estimated more accurately.

In summary, we obtain the total power of host $h_i$:

$$P_{h_i,t} = P_{i,t}^{cpu} + P_{i,t}^{mem} + P_{i,t}^{base} \tag{24}$$

Combining Equation (24) into Equation (1), we obtain the energy consumption cost $C_H$ of all hosts. We divide the life cycle of the CDC into multiple time segments and analyze the energy consumption separately in each time segment. Then, we have:

$$C_H = EP \times \sum_{k=1}^{L} \int_0^T \lambda_{i,k} \times P_{h_i,t} dt. \tag{25}$$

### 3.2. VM Migration Cost

We assume that at the beginning of each time segment, the CDC performs server consolidation to achieve the balance between the CSP's cost and the user's performance. VM migration is an important part of server consolidation. In a cluster composed of multi-core processor hosts, there are two types of VM migrations. The first is the inter-core migration on the same host, and the second is the inter-host migration between different hosts. Inter-core migration occurs when the core where the VMs are located is overloaded, and other cores of the same processor have sufficient computing resources. The VM migrates from one core of the processor to another core in a very short period of time through inter-core or inter-die communication. The inter-core migration does not involve memory, and the main impact is the hit rate of the processor cache. Therefore, the energy overhead of inter-core migration is negligible.

Next, we discuss the energy cost of inter-host migration. We use live migration technology to migrate VMs between different hosts. During live migration of a VM, the memory data of the VM is transmitted. Although VMs generate dirty pages during live migration, the research [28] indicates that the energy consumption of a VM live migration is positively related to the memory size of that VM. Therefore, we can assume that the larger the VM memory size is , the longer the migration time and the more energy consumption will be.

When migrating a VM $v_j$ from host $h_i$ to another host $h'_i$, we assume that $h_i$ reserves enough resources to support the migration of $v_j$, and $h_{i'}$ also reserves enough resources to run $v_j$. Buyya et al. [20] assumed that a VM would consume an extra 10% CPU usage to maintain the migration. In this paper, we extend this assumption to the memory resource usage of VM migration. In addition, we assume that the CDC deploys an exclusive network for VM migrations. We denote the size of the dedicated migration bandwidth of $h_i$ as $MIG\_NET_i$. The total cost of VM migrations in a given life cycle is denoted as $C_{mig}$. $C_{mig}$ is described as:

$$C_{mig} = \sum_{k=1}^{L} (C_k^{mig\_cpu} + C_k^{mig\_mem}),$$ (26)

where $C_k^{mig\_cpu}$ and $C_k^{mig\_mem}$ are the migration costs caused by CPU and memory in $t_k$, respectively.

$C_k^{mig\_mem}$ is calculated as:

$$C_k^{mig\_mem} = \sum_{i=1}^{N} \sum_{j=1}^{M} [EP \times \int_{t=0}^{t_{j,k}^{mig}} (\gamma_{j,i,x_i,i',x_{i'},k} \times P_{j,k}^{mig\_mem}) dk],$$ (27)

where $\gamma_{j,i,x_i,i',x_{i'},k}$ is a 0-1 indicator, $P_{j,k}^{mig\_mem}$ is the power consumption generated by migrating the memory data of $v_j$, and $t_{j,k}^{mig}$ is the time spent migrating $v_j$. If VM $v_j$ needs to be migrated from the $x_i$-th core of the processor of the host $h_i$ to the $x_{i'}$-th core of another host $h_{i'}$, then $\gamma_{j,i,x_i,i',x_{i'},k} = 1$; otherwise $\gamma_{j,i,x_i,i',x_{i'},k} = 0$. Since VM memory is the main data transferred during migration, we have:

$$t_{j,k}^{mig} = \frac{s_{j,k}^{mem}}{mig\_bw_{i,k}},$$ (28)

where $mig\_bw_{i,k}$ is the migration bandwidth size assigned to $v_j$. We consider that the migration bandwidth is evenly assigned to every migrated VM within $t_k$ on $h_i$. Hence, for a given source host $h_i$ and a destination host $h_{i'}$, we obtain:

$$mig\_bw_{i,k} = \frac{MIG\_NET_i}{\sum_{i=1}^{N} \sum_{j=1}^{M} \sum_{i'=1}^{N} \gamma_{j,i,x_i,i',x_{i'},k}},$$ (29)

then we have

$$t_{j,k}^{mig} = \frac{s_{j,k}^{mem} \times \sum_{i=1}^{N} \sum_{j=1}^{M} \sum_{i'=1}^{N} \gamma_{j,i,x_i,i',x_{i'},k}}{mig\_bw_{i,k} \times MIG\_NET_i}.$$ (30)

After this, we substitute Equation (30) into Equation (27). We let $p_{j,k}^{vmem}$ be the memory power of $v_j$ within $t_k$, and the memory migration cost of $v_j$ is $0.1 \times p_{j,k}^{vmem} = 0.1 \times \alpha_i^{mem} \times s_{j,k}^{mem}$.

Next, we discuss $C_k^{mig\_cpu}$. We assume here that the power consumption generated by a host in a CDC is mainly used to keep the VM running. Since the processor power consumption $P_{i,k}^{cpu}$ is related to its respective core in the current utilization ($c_i^{core_{cn_i}} - r_i^{core_1}, c_i^{core_{cn_i}} - r_i^{core_2}, \cdots, c_i^{core_{cn_i}} - r_i^{core_{cn_i}}$), it can be written as $P_{i,k}^{cpu}(c_i^{core_{cn_i}} - r_{i,k}^{core_1}, c_i^{core_{cn_i}} - r_{i,k}^{core_2}, \cdots, c_i^{core_{cn_i}} - r_{i,k}^{core_{cn_i}})$. For a given core on the processor $core_x$, where $x \in [1, cn_i]$, if a VM needs to be migrated to another host at this time, its CPU utilization $u_{i,k}^{mig\_core_x}$ is:

$$u_{i,k}^{mig\_core_x} = c_i^{core_x} - r_{i,k}^{core_x} + 0.1 \times \sum_{j=1}^{M} (\gamma_{j,i,x_i,i',x_{i'},k} \times s_{j,k}^{cpu}).$$ (31)

Hence, the power consumption of host $h_i$ during inter-host migration is:

$$(P_{i,k}^{cpu})' = P_{i,k}^{cpu}(u_{i,k}^{mig\_core_1}, \cdots, u_{i,k}^{mig\_core_{cn_i}}). \tag{32}$$

Then, we combine Equation (32) into Equation (2). We denote the updated host energy consumption cost $C_H$ as $C_H'$.

SLAV Penalty Cost

In a CDC, to guarantee user QoS, CSPs must provide SLAV compensation to relevant users in some form. This part of the overhead needs to be included in the cost consideration of the CDC. In this paper, we extend the single-core CPU SLAV definition by Buyya et al. [20] to multi-core CPU and memory. They are denoted as $SLAV_{cpu}$ and $SLAV_{mem}$, respectively.

For the processor, it is considered overloaded only if all its cores are overloaded. Hence, we have

$$SLAV_{cpu} = \frac{1}{N}\sum_{i=1}^{N}\frac{T_i^{s,cpu}}{T_i^{a,cpu}} \times \frac{1}{M}\sum_{j=1}^{M}\sum_{k=1}^{L}\frac{ud_{i,k}^{d,cpu}}{s_{i,k}^{r,cpu}}, \tag{33}$$

where $T_i^{s,cpu}$ is CPU SLAV duration caused by all cores overloaded on $h_i$, $T_i^{a,cpu}$ is the total working duration of the host, and $d_i^{d,cpu}$ is the size of the unsatisfied CPU resource demand as a result of $v_j$ migration in $t_k$.

Likewise, we propose the formal definition of $SLAV_{mem}$:

$$SLAV_{mem} = \frac{1}{N}\sum_{i=1}^{N}\frac{T_i^{s,mem}}{T_i^{a,mem}} \times \frac{1}{M}\sum_{j=1}^{M}\sum_{k=1}^{L}\frac{ud_{i,k}^{d,mem}}{s_{i,k}^{r,mem}}. \tag{34}$$

We denote CPU and memory SLAV compensation price indices as $pun_{cpu}$ and $pun_{mem}$, respectively. Then, we have:

$$C_{SLAV} = pun_{cpu} \times SLAV_{cpu} + pun_{mem} \times SLAV_{mem}. \tag{35}$$

*3.3. Problem Description*

In the above Section 3.1, we analyze the factors involved in the operating cost in a CDC, which are the host energy consumption cost $C_H'$, the VM migration cost $C_{mig}$, and the SLAV penalty cost $C_{SLAV}$. In this paper, our research goal is to minimize the associated operating cost $C$ of the CDC. Combining the above models, we have a minimizing multi-core-host-based cost problem in server consolidation (MMCC):

$$MIN \quad C = C_H' + \sum_{k=1}^{L} C_k^{mig\_mem} + C_{SLAV}. \tag{36}$$

A 0–1 indicator $\beta_{i,j,x_i,k}$ is used to mark whether the VM $v_j$ is running on the $x_i$-th core of the host $h_i$'s processor at the beginning of the $t_k$ time period. If $v_j$ runs on the $x_i$-th core of the host $h_i$, then $\beta_{i,j,x_i,k} = 1$, otherwise $\beta_{i,j,x_i,k} = 0$. The constraints of the MMCC problem are:

$$\sum_{i=1}^{N}\sum_{x_i=1}^{core_i} \beta_{i,j,x_i,k} = 1, \forall j, \forall k, \tag{37}$$

$$\sum_{i'=1}^{N}\sum_{x_{i'}=1}^{core_{i'}} \gamma_{j,i,x_i,i',x_{i'},k} = 1, \quad i \neq i', \forall j, \forall k, \tag{38}$$

$$\sum_{j=1}^{M} \beta_{i,j,x_i,k} \times s_{j,k}^{cpu} \leq r_{i,k}^{core_i}, \forall i, \forall x_i, \forall k, \tag{39}$$

$$\sum_{j=1}^{M} \beta_{i,j,x_i,k} \times s_{j,k}^{mem} \leq r_{i,k}^{mem}, \forall i, \forall k. \tag{40}$$

Constraint (37) means that in any period, any VM can only run on a specific core on a unique specific host. Constraint (38) means that in any period, a VM migrated from any host can only have a unique destination host and a unique core. Constraint (39) and (40) mean that in any period, the CPU and memory resources provided by each host to the VM cannot exceed its resource upper limits.

In the following, we analyze the complexity of the MMCC problem by considering a simple case of the problem. If the hosts in the CDC are homogeneous, the resource requirements of any VM $v_j$ in any time segment $t_k$ are fixed values and satisfy constraints (39) and (40). Then, the VM migration cost and SLAV penalty cost are both zero, and the objective function of the MMCC problem is:

$$MIN \quad C = C_H. \tag{41}$$

Obviously, the MMCC problem in this simple case can be reduced to the bin-packing problem. Since the bin-packing problem is NP-hard, the MMCC problem is also NP-hard.

## 4. Solution for MMCC Problem

Since the MMCC problem is NP-hard, we propose a heuristic based on the traditional four-step method for dealing with server consolidation. The first step is host workload detection, the second step is VM selection, the third and fourth steps are VM placements for VM from the overloaded and underloaded hosts. Before performing host overloading detection and VM selection, we will first predict the future workload trends of the VM based on its workload history. The purpose of this is to balance the load of hosts before they become overloaded and trigger SLAV occurrences, thereby reducing costs as much as possible.

### 4.1. VM Workload Prediction

Before predicting the future workload of a VM, we first need to preprocess its workload trace. The sampling frequency and precision cause a certain deviation between the historical sampling records and the actual usage of resources by the VM. To minimize the impact of these biases on the final result, we denoise by assuming that there is noise in the workload's history. In addition, we do not need to spend high computing power and a lot of time to obtain accurate prediction results. We only need to roughly judge a general trend of the VM's resource usage in the future.

In this paper, we utilize a classic denoise autoencoder [36] (DAE) based filter algorithm to preprocess the workload of VMs. Figure 2 shows the general structure of the DEA mechanism.

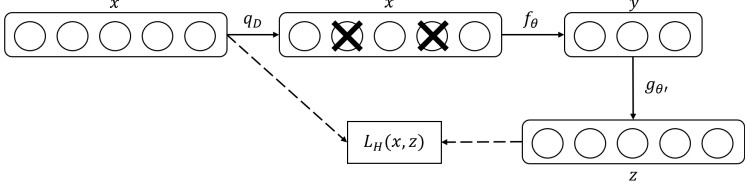

**Figure 2.** Denoise autoencoder.

In Figure 2, $x$ is the initialization input, and $\tilde{x} \sim q_D(\tilde{x}|x)$ is the stochastic mapping of $x$. Then, the autoencoder maps $\tilde{x}$ to $y$ with the encoder $f_\theta$ and generates the reconstruction $z$ with the decoder $g_{\theta'}$. The reconstruction error is measured by the loss function $L_H(x,z)$. In our proposed DAE-based filter, three autoencoders and one compression decoder are assembled, and their network structures are shown in Figures 3–6. Figure 7 shows the result of processing a segment of CPU usage records of a VM using the DEA-based filter.

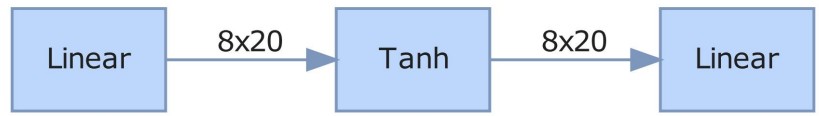

**Figure 3.** The network structure of the first autoencoder of the DAE-based filter.

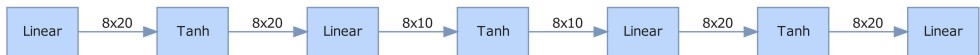

**Figure 4.** The network structure of the second autoencoder of the DAE-based filter.

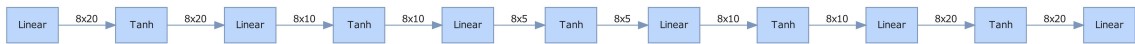

**Figure 5.** The network structure of the third autoencoder of the DAE-based filter.

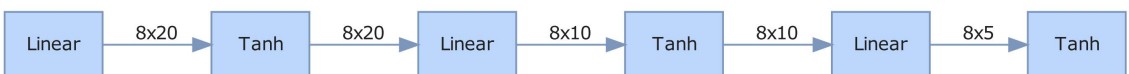

**Figure 6.** The network structure of the compression decoder of the DAE-based filter.

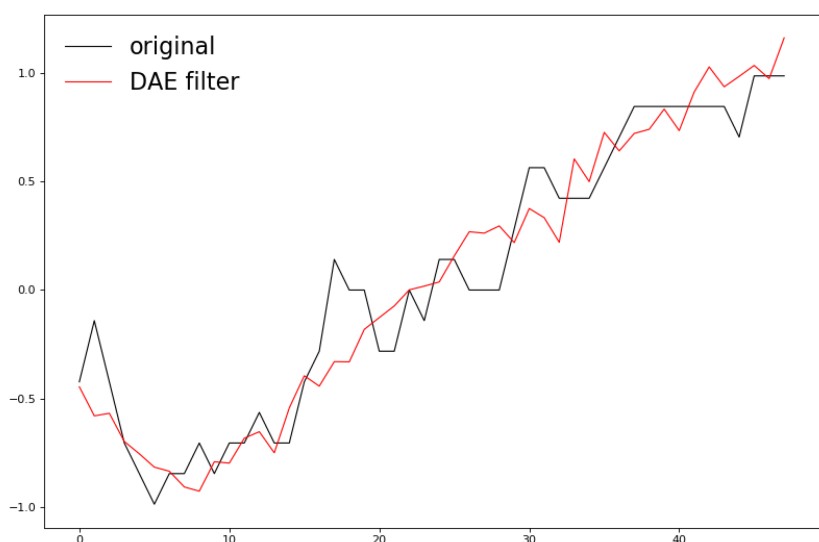

**Figure 7.** Example of the DAE-based filter.

Traditional RNNs cannot be parallelized, so there is a problem of slow training speed. To address this issue, we employ an SRU-based approach to predict the workload of VMs. Simple recurrent units (SRU) [37] eliminate the time dependency of most operations, enabling parallel processing. Experiments [37] show that the processing speed of the SRU is more than ten times faster than that of traditional LSTM under the condition of similar result accuracy. Since the SRU has been open-sourced and its usage method is not much different from LSTM, we will not discuss the theoretical details of SRU in this article.

After predicting the resource usage of each VM at the next time segment, we can perform host load detection and VM selection.

### 4.2. Host Workload Detection

The purpose of host overloaded detection is to avoid and eliminate the fierce competition of VMs for resources, thereby reducing the occurrences of SLAV. Common host overloaded detection methods are divided into two categories, static threshold method and dynamic threshold method. In the static threshold method, the resource uasge thresholds are set as fixed values. When the usage exceeds the threshold, the host is in an overloaded state, and SLAV occurs. At this time, the VMs must be migrated to reduce the load. In the dynamic threshold, CSPs analyze the use of computing resources through various statistical

methods to determine whether the competition for resources is fierce and whether the hosts are overloaded. The advantage of the static threshold method is that the host resources are fully utilized, but the disadvantage is that more overhead is required to reduce the SLAV. The advantage of the dynamic threshold method is that it can effectively reduce the SLAV, but the disadvantage is that sometimes the usages of hosts' resources are not sufficient. Therefore, we combine the advantages of the two and propose the double insurance-based fixed threshold overloading detection method (DIFT).

In DIFT, the first insurance is that the host cannot overload the CPU and memory resources during the current period. The second insurance is that the host cannot overload the CPU and memory resources in the next period. For a given host $h_i$, DIFT first detects whether the usages of various resources on $h_i$ exceed the given thresholds at the beginning of the $t_k$ time period, and then, based on the prediction results of the SRU method, we judge in the next time period $t_{k+1}$ whether the usages of various resources on $h_i$ exceed the given thresholds.

Since the VM migrations are divided into inter-core migrations and inter-host migrations, we correspondingly divide the CPU overload of the host into two situations: processor-overloaded and cores-overloaded. When the host is processor-overloaded, all cores on the processor are in an overloaded state. When the host is cores-overloaded, some (but not all) cores on the processor are in the overloaded state.

Let the overloaded threshold be $TH_{up} = TH_{up}^{cpu}, TH_{up}^{mem}$, where both $TH_{up}^{cpu}$ and $TH_{up}^{mem}$ are in the interval $(0, 1)$. For any $core_{x_i}$ on $h_i$, when the following inequality holds in $t_k$, it is in the state of processor-overloaded:

$$\sum_{j=1}^{M} \beta_{i,j,x_i,k} \times s_{j,k}^{cpu} > TH_{up}^{cpu} \times c_i^{core_{x_i}}, \tag{42}$$

$$\sum_{j=1}^{M} \beta_{i,j,x_i,k} \times s_{j,k+1}^{cpu} > TH_{up}^{cpu} \times c_i^{core_{x_i}}. \tag{43}$$

For some $core_{x_i}$ on $h_i$ (the number of $core_{x_i}$ that satisfy the condition cannot exceed $cn_i$), when the above in Equations (42) and (43) are established in $t_k$, it is in the cores-overloaded state.

Host $h_i$ is in a memory-overloaded state when the following inequality holds in $t_k$:

$$\sum_{j=1}^{M} \beta_{i,j,x_i,k} \times s_{j,k}^{mem} > TH_{up}^{mem} \times c_i^{mem}, \tag{44}$$

$$\sum_{j=1}^{M} \beta_{i,j,x_i,k} \times s_{j,k+1}^{mem} > TH_{up}^{mem} \times c_i^{mem}. \tag{45}$$

When the host is memory-overloaded or processor-overloaded, it must be in the host-overloaded state, and VM inter-host migration is required at this time. The situation where the host has only cores-overloaded is called semi-overloaded, and inter-core migrations can be preferentially leveraged at this time.

For an underloaded host, all VMs on it are migrated to other suitable hosts through inter-host migration; hence there is no need to consider inter-core migration requirements. Let the underloaded threshold be $TH_{down} = TH_{down}^{cpu}, TH_{down}^{mem}$, where both $TH_{down}^{cpu}$ and $TH_{down}^{mem}$ are in the interval $(0, 1)$. For $core_{x_i}$ on $h_i$, when the following inequalities hold in $t_k$, it is in the host-underloaded state:

$$\sum_{j=1}^{M} \beta_{i,j,x_i,k} \times s_{j,k}^{cpu} < TH_{down}^{cpu} \times c_i^{core_{x_i}}, \tag{46}$$

$$\sum_{j=1}^{M} \beta_{i,j,x_i,k} \times s_{j,k+1}^{cpu} < TH_{down}^{cpu} \times c_i^{core_{x_i}}. \tag{47}$$

$$\sum_{j=1}^{M} \beta_{i,j,x_i,k} \times s_{j,k}^{mem} < TH_{down}^{mem} \times c_i^{mem}, \tag{48}$$

$$\sum_{j=1}^{M} \beta_{i,j,x_i,k} \times s_{j,k+1}^{mem} < TH_{down}^{mem} \times c_i^{mem}. \tag{49}$$

### 4.3. VM Selection

VM selection is for overloaded hosts. The reason why we use the DIFT method is to avoid host overload and SLAV as much as possible, rather than react passively after SLAV occurs. Therefore, we can assume that in the $t_{k+1}$ time segment, there would be slight SLAV and host overload in the CDC. Our priority in VM selection is to select the VMs on each host $h_i$ that may cause $h_i$ to be overloaded during $t_{(k+1)}$ at the $t_k$ time segment and form a list of VMs to be migrated. If after the migrations of these VMs are completed, $h_i$ is still in the overloaded state during the $t_k$ period, then targeted processing will be performed. We discuss VM selection strategies under various overloaded states within $t_l$ (*e.g.*, $l = k + 1$) in different cases.

- Case 1: Host with semi-overloaded

In this case, we need to reduce the load on each overloaded core. Given the $j$-th overloaded core $core_{i,j,l}$ on host $h_i$ in $t_l$, we denote the set of $n$ VMs running on it as $V_{i,j,k} = \{v_{i,j,k}^1, v_{i,j,k}^2, \ldots, v_{i,j,k}^n\}$, its total resources are $c_{core_{i,j,k}}$, and the current available resource is $r_{core_{i,j,k}}$. For a VM $v_{i,j,k}^q \in V_{i,j,k}$, the amount of CPU resources it uses is denoted as $cpu_{v_{i,j,k}^q}$. Then, each selection chooses the VM $v_{i,j,k}^q$ that has the minimum value of $|(1 - TH_{up}^{cpu}) \times c_{core_{i,j,k}} - (r_{core_{i,j,k}} + cpu_{v_{i,j,k}^q})|$ into the inter-core migration list. We select a VM at a time until $r_{core_{i,j,k}} \geq (1 - TH_{up}^{cpu}) \times c_{core_{i,j,k}}$.

- Case 2: Host with only memory overloaded

Given a memory-overloaded host $h_i$ in $t_l$, we denote the set of $n$ VMs running on it as $V_{i,j,k} = \{v_{i,j,k}^1, v_{i,j,k}^2, \ldots, v_{i,j,k}^n\}$, the total amount of memory resource it has is $c_{mem_{i,l}}$, and the currently available amount of resources is $r_{mem_{i,l}}$. For a VM $v_{i,k}^q \in V_{i,k}$, the amount of memory resources used by it is recorded as $mem_{v_{i,l}^q}$. Then, each selection chooses the VM $v_{i,k}^q$ that has the minimum value of $|(1 - TH_{up}^{mem}) \times c_{mem_{i,k}} - (r_{mem_{i,k}} + mem_{v_{i,k}^q})|$ into the inter-host migration list. We select a VM at a time until $r_{mem_{i,k}} \geq (1 - TH_{up}^{mem}) \times c_{mem_{i,k}}$.

- Case 3: Host with only processor overloaded

We select VMs from each core in the same method as proposed in Case 1. All selected VMs are added into the inter-host migration list.

- Case 4: Host with processor overloaded or cores overloaded and memory overloaded

We first use the method in Case 1 to select VMs from each overloaded core. After the load of all cores drops under the overloaded threshold, if the memory load also drops under the overload threshold, the VM selection is completed; otherwise, the method in Case 2 is used to select VMs to reduce the memory load. All selected VMs are put into the inter-host migration list.

For a given overloaded host, at the beginning of the $t_k$ time segment, the above VM selection strategies are executed for its overloaded condition in $t_{k+1}$. If the host is still in an overloaded state in the $t_k$ time segment, the above strategies are executed again to reduce the current load.

### 4.4. VM Placement

To make full use of the resources of the hosts, we should fully consider the space and time competition of different VMs for different resources when placing VMs on hosts.

In the VM selection phase, we obtain the inter-core migration list and inter-host migration list. Regarding a semi-overloaded host, it should be noted that the load of the cores may not be reduced through inter-core migration. Therefore, in the VM placement phase, we first process the inter-core migration list and then add the remaining VMs to the inter-host migration list to process together.

There are two goals of VM placement: (1) to ensure that the resources of the target host can be fully utilized during the $t_k$ period; and (2) after the VM is placed on the destination host $h_i$, the host will not be in the overloaded state during the $t_{k+1}$ period.

We address the inter-core migration list first. For the a semi-overloaded host $h_i$, we sort all un-overloaded cores $\{core_1, \cdots, core_{cn'_i}\}$ in ascending order of load, where $cn'_i$ is the number of overloaded cores. We denote this orderd sequence as $ordered\_uo\_cores_{i,k}$. We arrange the VMs in the inter-core migration list $icm_i$ of $h_i$ in descending order according to the current demand for CPU resources to form the list $ordered\_icm_i$. We take the first VM from $ordered\_icm_i$ and traverse $ordered\_uo\_cores_{i,k}$ for it in order to find the first core with enough CPU resources. If a suitable core cannot be found in $ordered\_uo\_cores_{i,k}$ for this VM, we add it to the inter-host migration list. The VM is then removed from $ordered\_icm_i$. We repeat the above operations until $ordered\_icm_i$ is empty. Each semi-overloaded host needs to execute this placement algorithm for its VMs in $icm_i$. Algorithm 1 demonstrates the pseudocode of the inter-core VM placement algorithm.

---

**Algorithm 1** Inter-Core VM Placement Algorithm.

---

**Input:** host $h_i$, inter-core migration list $icm_i$ of $h_i$
**Output:** allocation of VMs on certain cores
1: $Get\_sorted(core_1, \cdots, core_{cn'_i}) \rightarrow ordered\_uo\_cores_{i,k}$
2: $Get\_sorted(icm_i) \rightarrow ordered\_icm_i$
3: **for** each $vm_j$ in $ordered\_icm_i$ **do**
4:     **for** each $core_p$ in $ordered\_uo\_cores_{i,k}$ **do**
5:         **if** $core_p$ is available for $vm_j$ in $t_k$ and $t_{k+1}$ **then**
6:             $allocation.add(vm_j, h_j.core_p)$
7:             $ordered\_icm_i.remove(vm_j)$
8:         **end if**
9:         break
10:     **end for**
11: **end forreturn** allocation

---

Next, the inter-host migration list is processed. First, all the hosts are divided into two categories according to the intensity of CPU and memory usage: CPU-intensive hosts and memory-intensive hosts. The following calculation method is used to classify a given host $h_i$. We take the workload trace of $h_i$ in 12 consecutive time segments (one hour), where the normalized CPU workload time series is $LD_{i,k}^{cpu} = \{ld_{i,k-10}^{cpu}, \cdots, ld_{i,k}^{cpu}, ld_{i,k+1}^{cpu}\}$, and the time series of normalized memory workload is $LD_{i,k}^{mem} = \{ld_{i,k-10}^{mem}, \cdots, ld_{i,k}^{mem}, ld_{i,k+1}^{mem}\}$. Since $h_i$ has a multi-core CPU, its normalized CPU workload at period $t_k$ is:

$$ld_{i,k}^{cpu} = \prod_{x=1}^{cn_i} \frac{c_i^{core_{cn_i}} - r_{i,k}^{core_x}}{max\{c_i^{core_{cn_i}} | i \in [1, N]\}}. \tag{50}$$

The reason why the denominator is $max\{c_i^{core_{cn_i}} | i \in [1, N]\}$ is that CPUs with different performances can be compared with each other through normalization. The smaller the value of $ld_{i,k}^{cpu}$ is, the lower the CPU load of $h_i$ in the time period $t_k$.

At a certain time period $t_k$, its normalized memory workload is:

$$ld_{i,k}^{mem} = \frac{c_i^{mem} - r_{i,k}^{mem}}{max\{c_i^{mem} | i \in [1, N]\}}. \tag{51}$$

The smaller the value of $ld_{i,k}^{mem}$ is, the lower the memory load of $h_i$ in the time period $t_k$.

Based on Equation (50), we calculate the CPU score of $h_i$:

$$score_{i,k}^{cpu} = \frac{1}{10}(\sum_{y=1}^{12} ld_{i,k}^{cpu} - max(LD_{i,k}^{cpu}) - min(LD_{i,k}^{cpu})), \tag{52}$$

where $max(LD_{i,k}^{cpu})$ is the maximum value in the normalized CPU workload sequence, and $min(LD_{i,k}^{cpu})$ is the minimum value in the normalized CPU workload sequence. We remove $max(LD_{i,k}^{cpu})$ and $min(LD_{i,k}^{cpu})$ from $\sum_{y=1}^{12} ld_{i,k}^{cpu}$ to minimize the impact of possible severe load fluctuations on the score.

Based on Equation (51), we calculate the memory score of $h_i$:

$$score_{i,k}^{mem} = \frac{1}{10}(\sum_{y=1}^{12} ld_{i,k}^{mem} - max(LD_{i,k}^{mem}) - min(LD_{i,k}^{mem})), \tag{53}$$

where $max(LD_{i,k}^{mem})$ is the maximum value in the normalized memory workload sequence, and $min(LD_{i,k}^{mem})$ is the minimum value in the normalized memory workload sequence.

If $score_{i,k}^{cpu} \geq score_{i,k}^{mem}$, $h_i$ is CPU-intensive type; otherwise, $h_i$ is a memory-intensive type. The CPU-intensive type hosts have more abundant available memory resources, and the memory-intensive type hosts have more abundant available CPU resources. Therefore, in the time period $t_k$, the hosts of the CPU-intensive type are arranged in ascending order according to their values of $ld_{i,k}^{mem}$, forming a list *memordered_cpu_hosts_list*. Memory-intensive hosts are sorted in ascending order according to their $ld_{i,k}^{cpu}$ values to form a list *cpuordered_mem_hosts_list*. The reason for using the above sorting method is: CPU-intensive hosts have enough remaining memory resources, so VMs that require more memory resources can be placed on them; memory-intensive hosts have enough remaining CPU resources, so they can be placed with VMs that require more CPU resources.

In the following, we sort the VMs in the inter-host migration list by their resource usage requirements. The VMs to be migrated are also divided into CPU-intensive type and memory-intensive type. The CPU-intensive type VMs need to be placed on the memory-intensive type hosts as much as possible, and the memory-intensive VMs need to be placed on the CPU-intensive type hosts as much as possible. We use the following calculation method to classify a given VM $v_j$. We take the workload trace of $v_j$ in 12 consecutive time segments (one hour), where the normalized CPU workload time series is $VLD_{j,k}^{cpu} = \{vld_{j,k-10}^{cpu}, \cdots, vld_{j,k}^{cpu}, vld_{j,k+1}^{cpu}\}$, and the time series of normalized memory workload is $VLD_{j,k}^{mem} = \{vld_{j,k-10}^{mem}, \cdots, vld_{j,k}^{mem}, vld_{j,k+1}^{mem}\}$. At certain time period $t_l$, the normalized CPU workload of $v_j$ is:

$$vld_{j,k}^{cpu} = \frac{s_{j,k}^{cpu} - min\{s_{x,k}^{cpu}|x \in [1, M]\}}{max\{s_{x,k}^{cpu}|x \in [1, M]\} - min\{s_{x,k}^{cpu}|x \in [1, M]\}}. \tag{54}$$

The smaller the value of $vld_{j,k}^{cpu}$, the lower the CPU demand of $v_j i$ in the time period $t_k$. At certain time period $t_l$, the normalized memory workload of $v_j$ is:

$$vld_{j,k}^{mem} = \frac{s_{j,k}^{mem} - min\{s_{x,k}^{mem}|x \in [1, M]\}}{max\{s_{x,k}^{mem}|x \in [1, M]\} - min\{s_{x,k}^{mem}|x \in [1, M]\}}. \tag{55}$$

Based on Equation (54), we calculate the CPU score of $v_j$:

$$vscore_{j,k}^{cpu} = \frac{1}{10}(\sum_{y=1}^{12} vld_{j,k}^{cpu} - max(VLD_{j,k}^{cpu}) - min(VLD_{j,k}^{cpu})), \tag{56}$$

where $max(VLD_{j,k}^{cpu})$ is the maximum value in the normalized CPU workload sequence of $v_j$, and $min(VLD_{j,k}^{cpu})$ is the minimum value in the normalized CPU workload sequence of $v_j$.

Based on Equation (55), we calculate the memory score of $v_j$:

$$vscore_{j,k}^{mem} = \frac{1}{10}(\sum_{y=1}^{12} vld_{j,k}^{mem} - max(VLD_{j,k}^{mem}) - min(VLD_{j,k}^{mem})), \tag{57}$$

where $max(VLD_{j,k}^{mem})$ is the maximum value in the normalized memory workload sequence of $v_j$, and $min(VLD_{j,k}^{mem})$ is the minimum value in the normalized memory workload sequence of $v_j$.

If $vscore_{j,k}^{cpu} \geq vscore_{j,k}^{mem}$, $v_j$ is a CPU-intensive type; otherwise, $v_j$ is a memory-intensive type.

In the time period $t_k$, CPU-intensive type VMs are arranged in descending order according to their $vld_{j,k}^{cpu}$ values, forming a list *ordered_cpu_vms_list*. We pick out the VMs in *ordered_cpu_vms_list* in turn, traverse the *cpuordered_mem_hosts_list*, and find the first host that can meet the resource requirements of the current VM and will not be overloaded at both $t_k$ and future $t_{k+1}$.

Since the hosts have multi-core CPUs, we design the following judgment when deciding which core of the host $h_i$ will be used by the VM $v_j$. We sort the cores in $h_i$'s processor in descending order by their available resources $r_{i,k}^{core_1}, r_{i,k}^{core_2}, \cdots, r_{i,k}^{core_{cn_i}}$, which constitute the sequence *ordered_cores_{i,k}*. Then, the VM $v_j$ will be preferentially placed on the front core in *ordered_cores_{i,k}* (and this core will also meet the resource requirements of $v_j$ in the $t_{k+1}$ time period).

In the time period $t_k$, memory-intensive type VMs are arranged in descending order according to their values of $vld_{j,k}^{mem}$, forming a list *ordered_mem_vms_list*. We pick out the VMs in the *ordered_mem_vms_list* in turn, traverse the *memordered_cpu_hosts_list*, and find the first host that can meet the resource requirements of the current VM and will not be overloaded at both $t_k$ and future $t_{k+1}$. On the current host, the same multi-core placement method is used for processing *ordered_cores_{i,k}*.

When the destination host is determined for a given VM to be migrated, this VM is removed from the inter-host migration list. Algorithm 2 demonstrates the pseudocode of the inter-host VM placement algorithm. After the above *ordered_mem_vms_list* and *ordered_cpu_vms_list* are traversed, and there are still VMs to be migrated, the first-fit method is used to find available hosts in the host list for them. If there are still VMs to be migrated, the hosts in the energy-saving state are powered on one by one until all the VMs to be migrated are assigned destination hosts.

After the above process, we perform underloaded host detection on the hosts in the CDC. If there are still underloaded hosts at this time, the VMs on the underloaded hosts are added to form a VM migration list, and Algorithm 2 is executed.

---

**Algorithm 2** Inter-Host VM Placement Algorithm.

---

**Input:** hostlist, inter-host migration list
**Output:** allocation of VMs

1: $Get\_classification(host) \rightarrow cpu\_intensive\_hosts_k, mem\_intensive\_hosts_k$
2: $Get\_classification(inter-hostmigrationlist) \rightarrow cpu\_intensive\_vms_k,$
$\qquad mem\_intensive\_vms_k$
3: $Get\_sorted(cpu\_intensive\_hosts_k) \rightarrow memordered\_cpu\_hosts\_list$
4: $Get\_sorted(mem\_intensive\_hosts_k) \rightarrow cpuordered\_mem\_hosts\_list$
5: $Get\_sorted(cpu\_intensive\_vms_k) \rightarrow ordered\_cpu\_vms\_list$
6: $Get\_sorted(mem\_intensive\_vms_k) \rightarrow ordered\_mem\_vms\_list$
7: **for** each $vm_j$ in $ordered\_cpu\_vms\_list$ **do**
8:     **for** each $h_i$ in $cpuordered\_mem\_hosts\_list$ **do**
9:         $Get\_sorted(core_1, \cdots, core_{cn_i}) \rightarrow ordered\_cores_{i,k}$
10:         **for** each $core_p$ in $ordered\_cores_{i,k}$ **do**
11:             **if** $core_p$ is available for $vm_j$ in $t_k$ and $t_{k+1}$ **then**
12:                 $allocation.add(vm_j, h_i.core_p)$
13:                 $ordered_cpu_vms_list.remove(vm_j)$
14:             **end if**
15:             break
16:         **end for**
17:     **end for**
18: **end for**
19: **for** each $vm_j$ in $ordered\_mem\_vms\_list$ **do**
20:     **for** each $h_i$ in $memordered\_cpu\_hosts\_list$ **do**
21:         $Get\_sorted(core_1, \cdots, core_{cn_i}) \rightarrow ordered\_cores_{i,k}$
22:         **for** each $core_p$ in $ordered\_cores_{i,k}$ **do**
23:             **if** $core_p$ is available for $vm_j$ in $t_k$ and $t_{k+1}$ **then**
24:                 $allocation.add(vm_j, h_i.core_p)$
25:                 $ordered_cpu_vms_list.remove(vm_j)$
26:             **end if**
27:             break
28:         **end for**
29:     **end for**
30: **end for** return allocation

---

## 5. Performance Evaluation

In this section, we evaluate the performance of our proposed solution, named MMCC, with a real VM workload trace-driven simulation.

### 5.1. Experiment Setup

According to the energy consumption analysis and statistics of the hosts by Basmadjian et al. [34], Minartz et al. [38], and Jin et al. [39], we simulated three types of hosts as $H_{large}$, $H_{medium}$, and $H_{small}$, respectively. Their resource parameters are shown in Table 1, the power parameters are shown in Tables 2 and 3, and the capacitances of different components of the processor are given in Table 4. The numbers of $H_{small}$ hosts, $H_{medium}$ hosts, and $H_{large}$ hosts in the simulated CDC are both 100.

The VM workload trace dataset is from the Alibaba CDC [33]. The VM traces in the dataset are recorded by sampling every five minutes. We selected 1000 VMs in one day (a total of 288 time segments) from the dataset to simulate the consumers' demands for cloud services. The simulation was implemented on CloudMatrix Lite [40]. The DAE-based filter and the SRU algorithm (the source code is available at https://github.com/asappresearch/sru accessed on 19 October 2022) was based on PyTorch [41].

We set the electricity price as $EP = 0.25\$/kWh$. The SLAV penalty is a static value $pun_{cpu} = pun_{mem} = 0.01\$$ [42]. The host should reserve an extra 10% resources for

migrations. Thereby, we set $TH_{up}^{cpu} = TH_{up}^{mem} = 0.9$. We also set $TH_{down}^{cpu} = TH_{down}^{mem} = TH_{down}^{disk} = TH_{down}^{net} = 0.1$.

**Table 1.** Resource parameters of the hosts.

| Host Type | CPU | Memory |
|:---:|:---:|:---:|
| $H_{large}$ | Intel Xeon CPU (16 cores) | 8 GB |
| $H_{medium}$ | Intel Xeon CPU (8 cores) | 6 GB |
| $H_{small}$ | Intel Xeon CPU (4 cores) | 4 GB |

**Table 2.** Base power of the hosts.

| Host Type | Base Value (kW) |
|:---:|:---:|
| $H_{large}$ | 108.2 |
| $H_{medium}$ | 103.8 |
| $H_{small}$ | 98.5 |

**Table 3.** Memory power parameters.

| Host Type | Value | Memory (kW) |
|:---:|:---:|:---:|
| $H_{large}$ | $p_{peak}$ | 0.21736 |
| | $p_{idle}$ | 0.17576 |
| $H_{medium}$ | $p_{peak}$ | 0.10868 |
| | $p_{idle}$ | 0.08788 |
| $H_{small}$ | $p_{peak}$ | 0.05434 |
| | $p_{idle}$ | 0.04394 |

**Table 4.** Capacitance of different components of the processor.

| Component | Capacitance |
|:---:|:---:|
| Chip-Level Mandatory | 0.103 |
| Die-Level Mandatory | 0.301 |
| On-chip Cache | 0.165 |
| Off-chip Cache | 3.759 |
| Inter-die | 0.595 |

We combined four overloading detection algorithms (MAD [20], IQR [20], and LR [20]), three VM selection algorithms (MMT [20,25,30], MC [20,43], and RS [20]), and one VM placement algorithm (PABFD [20]) as nine baseline methods to compare with our proposed solution MMCC. All the abovementioned workload detection and selection algorithms were initially designed for single-core; hence, we modified them to work in the multi-core hosts by seeing the capacity of the CPU as the sum of its capacities of the cores. Moreover, The PABFD placement algorithm and its corresponding energy consumption model only take into account the host's sinlge-CPU. Therefore, we modified it here to suit our multi-core (by randomly selecting a core in the CPU for the VM) and multi-resource scenario. The new PABFD placement algorithm is PABFDM, as shown in Algorithm 3 for the pseudocode.

---

**Algorithm 3** PABFDM algorithm.

---

**Input:** hostList, vmList
**Output:** allocation of the VMs

1: vmList.sortDecreasingUtilization()
2: **for** each VM *in* vmList **do**
3:     minPower ← MAX
4:     **for** host *in* hostList **do**
5:         **if** *no SLAV on this host and this host meets the CPU and memory resource requirement for* VM **then**
6:             power ← *estimatePower*(*host*, *VM*)
7:             **if** power<minPower **then**
8:                 minPower←power
9:             **end if**
10:         **end if**
11:     **end for**
12:     *allocation.add*(*VM*, *host.random*(*core*))
13: **end for**
        **return** allocation

---

*5.2. Evaluation*

The metrics involved in the evaluation are host energy consumption cost, SLAV penalty cost, and the number of VM migrations. Since the CPU cost of the VM migration energy consumption belongs to the hosts' energy consumption during calculation, we used the number of VM migrations to indirectly measure the migration cost.

Figure 8 shows the host energy consumption for each time slice of the day when all the methods are used to perform server consolidation. Figure 9 compares the total host energy consumption over the day when all the methods are used to perform server consolidation. From Figure 8, it can be seen that the host energy consumption generated by MMCC is less than that of the baseline methods in most of the time periods. From Figure 9, it can be seen that the total host energy consumption generated by MMCC in a day is about 10% less than that of LR-MMT (the best in the baseline methods) and is about 43.9% less than that of MAD-RS (the worst in the baseline methods). In a cluster composed of multi-core processor hosts, MMCC can effectively schedule tasks among multiple cores to optimize energy consumption.

The comparison of CPU and memory SLAV produced by all the methods in a day is shown in Figure 10. The CPU SLAV generated by MMCC is much smaller than that of the baseline methods. For example, MMCC produces about 54% less CPU SLAV than that of MAD-RS and about 39% less than that of LR-MMT. Likewise, the memory SLAV produced by MMCC is much smaller than that of the baseline methods. A comparison of the total SLAV cost produced by all methods in one day is shown in Figure 11. MMCC outperforms the baseline methods. For instance, the total cost generated by MMCC is about 51.7% less than that generated by IQR-RS (the worst of the baseline methods) and about 33.5% less than that generated by LR-MMT (the best of the baseline methods). It can be said that the traditional server consolidation method represented by the baseline methods do not perform well in a cluster composed of multi-core processor hosts, while MMCC can better handle this scenario.

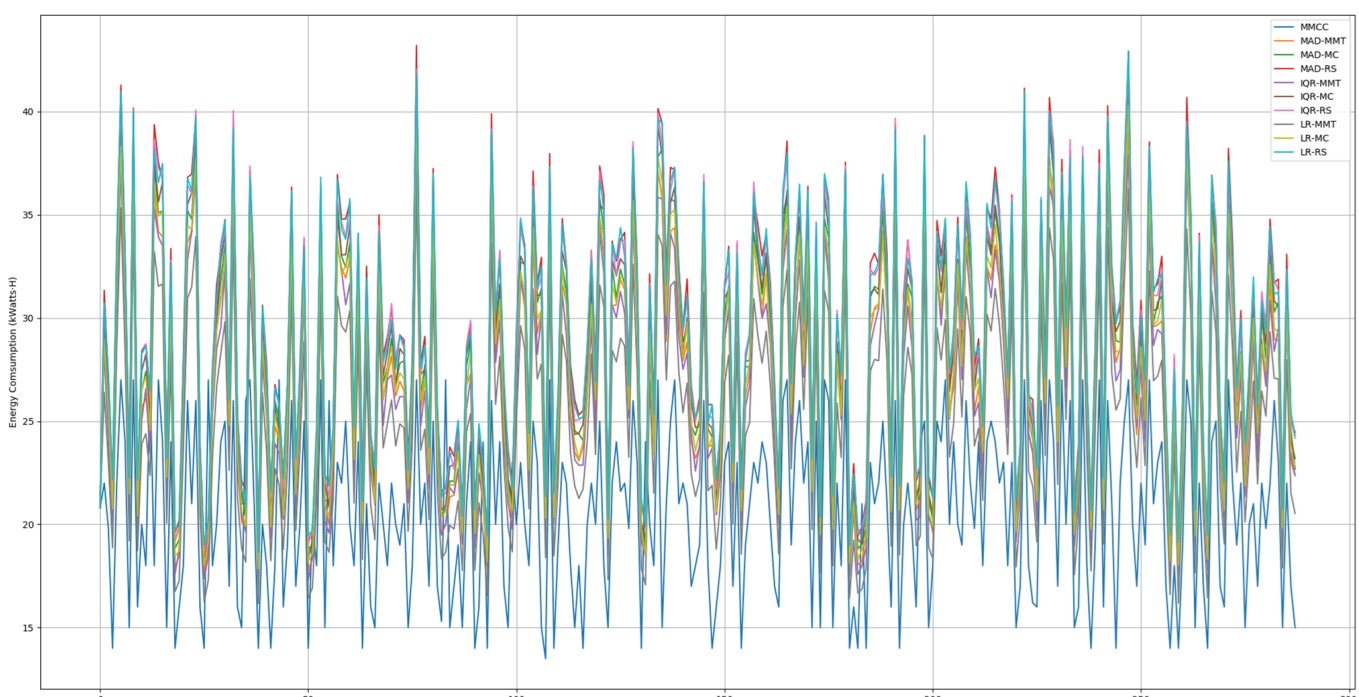

**Figure 8.** Comparing the energy consumption of hosts by all methods in every time segment.

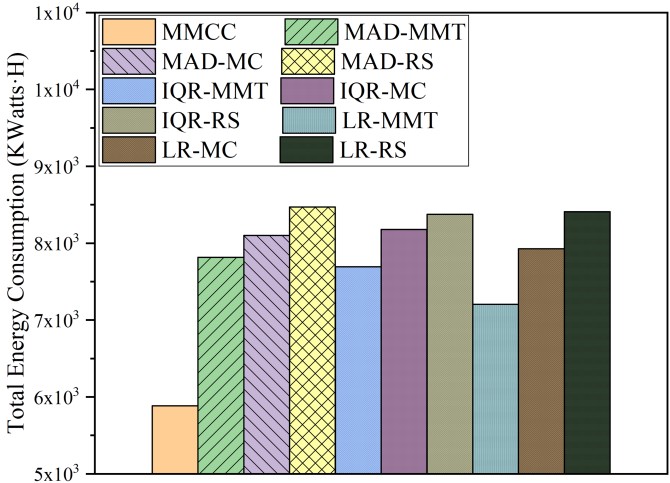

**Figure 9.** Comparing the total energy consumption hosts by all methods.

Figure 12 shows the number of VM migrations triggered in each time slice of the day when all the methods are used to perform server consolidation. Figure 13 compares the total number of VM migrations triggered in a day using all the methods to perform server consolidation. As can be seen from Figure 13, compared to the baseline method, the number of migrations triggered by MMCC does not have a large advantage. For example, MMCC triggers about 9.5% fewer migrations than that of IQR-RS. However, it should be noted that the VM migrations caused by MMCC in time $t_k$ is mainly to deal with the possible overloaded hosts in the future. Therefore, the SLAV produced by MMCC is much smaller than that of the baseline methods. In addition, part of the migrations caused by MMCC are inter-core migrations, which only happen inside the host. The cost of inter-core migration is negligible. The traditional baseline methods do not consider the inter-core migration in the case of multi-core processors.

Figure 14 shows and compares the total cost generated by all the methods in one day. MMCC outperforms the baseline methods. For instance, the total cost generated by

MMCC is about 20.9% less than that of LR-MMT (the best of the baseline methods) and about 34.4% less than that of MAD-RS (the worst of the baseline methods). MMCC can not only optimize the energy consumption in the environment of multi-core processor hosts, but also reduce the SLAV in server consolidation through the host load detection and VM selection strategies based on the prediction method.

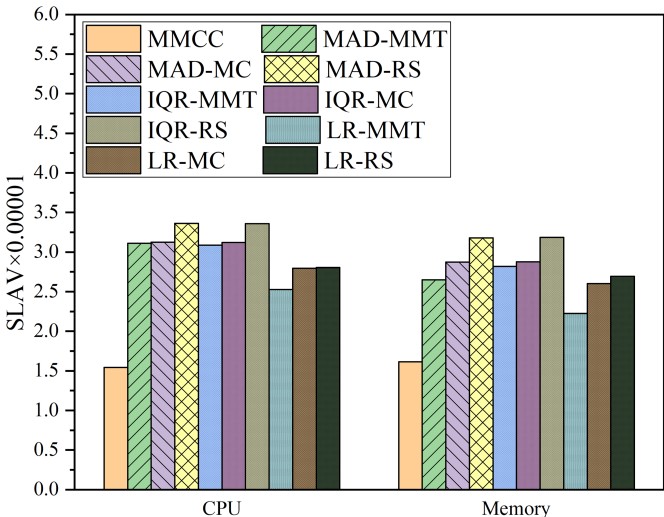

**Figure 10.** Comparing the SLAV by all methods regarding CPU and memory.

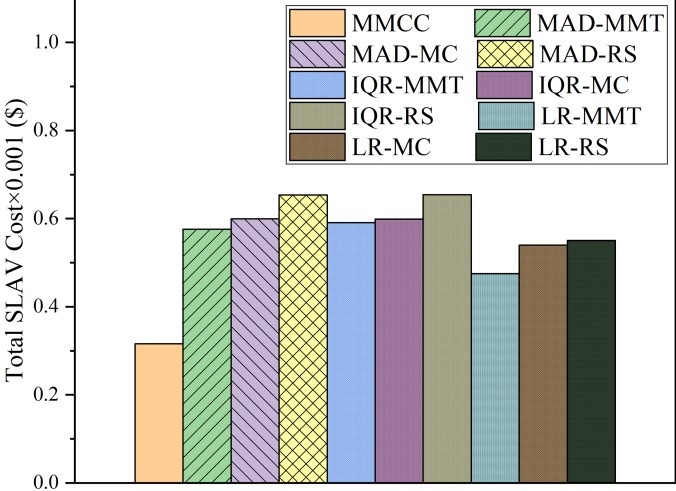

**Figure 11.** Comparing the total SLAV penalty cost by all methods.

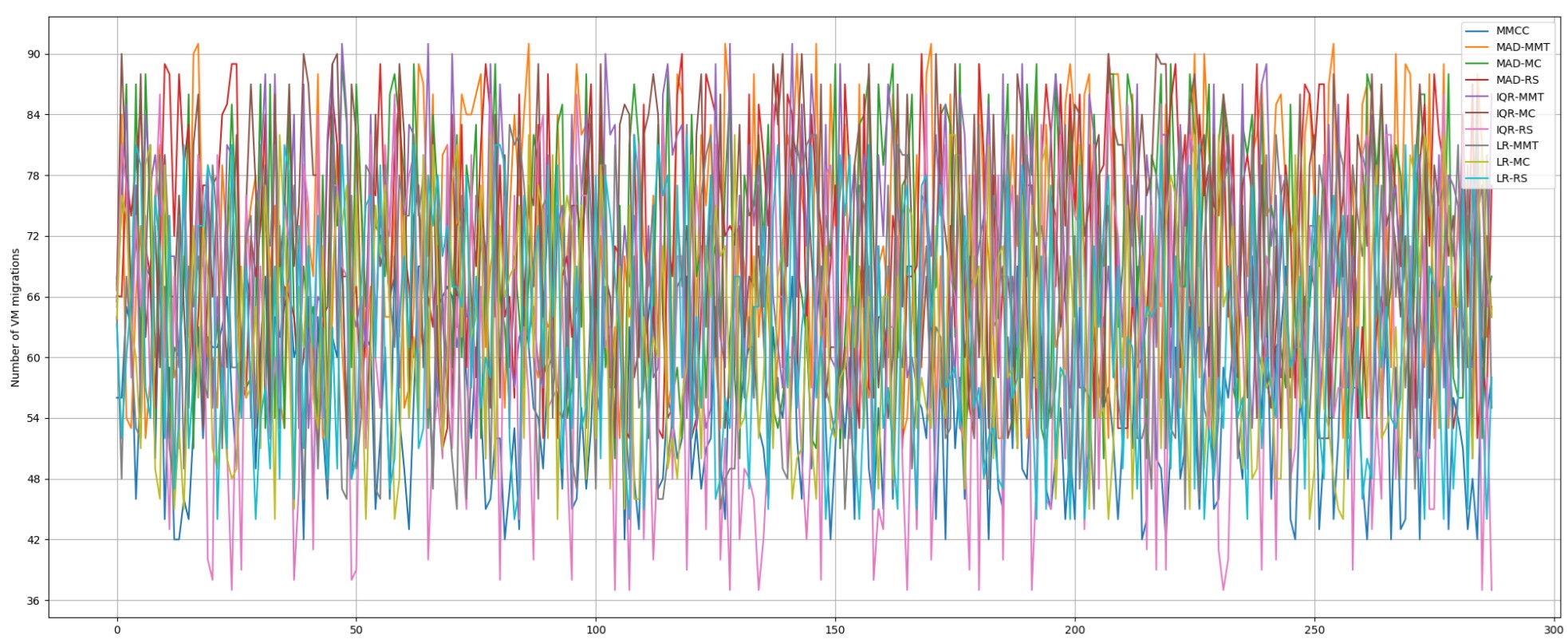

**Figure 12.** Comparing the number of VM migrations triggered by all methods in every time segment.

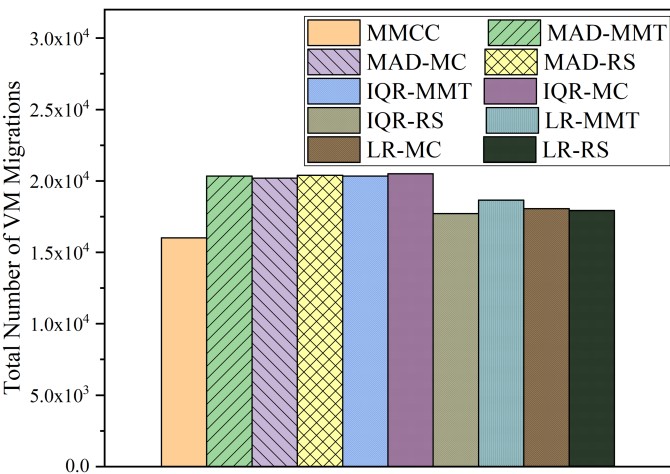

**Figure 13.** Comparing the total number of VM migrations triggered by all methods.

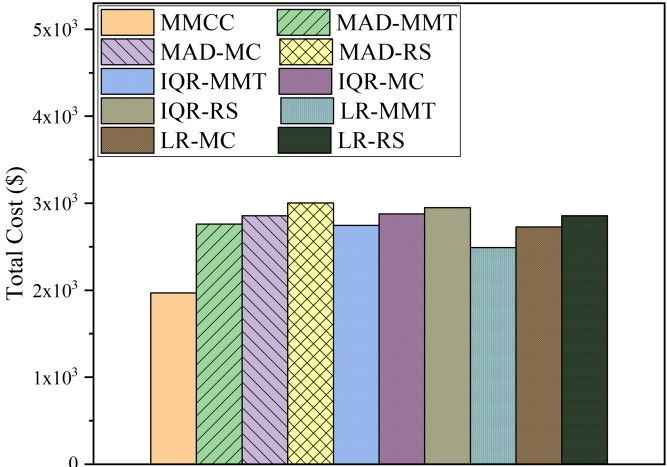

**Figure 14.** Comparing the total cost by all methods.

## 6. Conclusions

In this paper, we focus on reducing the total cost of server consolidation in a CDC, which is composed of multi-core processor hosts, operating costs while ensuring consumers' QoS. We established a cost model based on multi-core and multi-resource usage in the CDC, taking into account the host energy cost, VM migration cost, and SLAV penalty cost. Based on this model, we define the MMCC problem in server consolidation. We designed a heuristic solution to deal with this problem. We employ a DAE-based filter to preprocess the VM workload dataset and to reduce noise in the workload trace. Subsequently, an SRU-based method is used to predict the usage of computing resources, allowing us to trigger inter-core or inter-host VM migrations before the host enters the state. We design a muliti-core-aware heuristic algorithm to solve the VM placement problem. Finally, simulations driven by real VM workload traces verify the effectiveness of our proposed method. Compared with the existing server consolidation methods, our proposed MMCC can reduce host energy consumption from 10% to 43.9%, SLAV cost by 33.5% to 51.7%, and total cost by 20.9% to 34.4% in a multi-core hosts cluster.

In the future, we will first consider a more comprehensive cost model, such as taking into account the operational life span of the host, the network topology of CDC, and cooling system.

**Author Contributions:** Conceptualization, H.L. and Y.S.; methodology, H.L.; software, H.L.; validation, H.L., L.W., and Y.S.; formal analysis, H.L.; investigation, H.L. and Y.L.; resources, H.L. and Y.L.; data curation, H.L.; writing—original draft preparation, H.L., L.W., and Y.L.; writing—review and editing, H.L., L.W., Y.L., and Y.S.; visualization, H.L.; supervision, Y.S.; project administration, Y.S.; funding acquisition, H.L. and Y.S. All authors have read and agreed to the published version of the manuscript.

**Funding:** This research was funded by the National Natural Science Foundation of China (No.62002067), the Guangzhou Youth Talent Program (QT20220101174), the Department of Education of Guangdong Province (No.2020KTSCX039), Foundation of The Chinese Education Commission (22YJAZH091), and the SRP of Guangdong Education Dept (2019KZDZX1031).

**Institutional Review Board Statement:** Not applicable.

**Informed Consent Statement:** Not applicable.

**Data Availability Statement:** Not applicable.

**Conflicts of Interest:** The authors declare no conflict of interest.

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
