# Peer review of "More than Meets One Core: An Energy-Aware Cost Optimization in Dynamic Multi-Core Processor Server Consolidation for Cloud Data Center"

_electronics, doi:10.3390/electronics11203377_

Round 1
Reviewer 1 Report
The paper addresses an important challenge for the resource management in the Cloud data center by presenting a solution for Minimizing Multi-Core-host-based Cost problem in server consolidation.
The research is well structure, however its writing need to be proofread as there are a number of grammar and typo mistakes.
Please consider the following comments and suggestions:
- All abbreviations' details should be mentioned in the first appear of them.
- The paper's abstract starts with a short sentence about the Corona Virus Pandemic, where I don't think there are a logic reason to include it in the abstract.
- In the introduction section, the listed work motivation is not convincing, it needs to be more detailed. Also there are no clear goal nor contributions.
- Figures 3-6, the included text is unreadable
- Algorithm 2: lines (9,10,21,22) type mistake in "odered_cores"
- Figure 12 is unclear and the included text is unreadable
- The conclusions section, line 409: what "reducing the CDC" means? this should be clarified
Author Response
Response to Reviewer 1 Comments
Point 1:- All abbreviations' details should be mentioned in the first appear of them.
Response 1: Thank you very much for your suggestion.
We have given the full names of the abbreviations, such as:
Line 3: Quality of Service (QoS)
Line 6: Service level agreement violation (SLAV)
Line 28: cloud data centers (CDC)
Line 34: Service level agreement (SLA)
Line 36: Service level agreement violation (SLAV)
Line 35: Quality of Service (QoS)
Line 37: cloud service providers (CSPs)
Point 2:- The paper's abstract starts with a short sentence about the Corona Virus Pandemic, where I don't think there are a logic reason to include it in the abstract.
Response 2: Thank you very much for your suggestion. We have rewritten the abstract. Please check it below.
“The massive number of users has brought severe challenges in managing the cloud data centers (CDC) composed of multi-core processor hosts to cloud service providers. Guaranteeing the Quality of Service (QoS) of multiple users as reducing the operating cost of CDC is a major problem that needs to be solved. To solve this problem, this paper establishes a cost model based on multi-core hosts in CDC, which comprehensively considers the hosts' energy cost, virtual machine (VM) migration cost, and Service Level Agreement violation (SLAV) penalty cost. To optimize the goal, we design the following solution. We employ a DAE-based filter to preprocess the VM historical workload and use an SRU-based method to predict the computing resource usage of the VMs in future periods. Based on the predicted results, we trigger VM migrations before the hosts get into the overloaded state to reduce the occurrence of SLAV. A multi-core-aware heuristic algorithm is proposed to solve the placement problem. Simulations driven by the VM real workload dataset validate the effectiveness of our proposed method. Compared with the existing baseline methods, our proposed method reduces the total operating cost by 20.9~ 34.4%.”
Point 3:- In the introduction section, the listed work motivation is not convincing, it needs to be more detailed. Also there are no clear goal nor contributions.
Response 3: Thank you very much for your suggestion.
We have rewritten the introduction section from the second paragraph. Please check it below.
“Increasing the resource rate of cloud data centers (CDCs) is one of the most effective means to reduce management costs. But there is a conflict between reducing costs and the performance that cloud service customers get. To improve resource usage, virtual machines (VMs) or containers assigned to users must be highly concentrated on physical hosts. However, a high degree of centralization brings a high degree of resource competition. When the competition is too intense, the host may be overloaded, thereby reducing the performance and user experience of VMs. To ensure user experience, Service Level Agreement (SLA) is used to quantitatively describe the corresponding QoS. If the SLA cannot be maintained, the Quality of Service (QoS) is threatened and SLA violation (SLAV) is generated. When SLAV appears, cloud service providers (CSPs) need to provide compensation to users as punishment for failing to meet user performance requirements. Currently, server consolidation is used to dynamically adjust the load balance between hosts in a CDC. Server consolidation periodically checks the load of hosts in the cluster and initiates VM migration to achieve load balancing, thereby maintaining a balance between resource utilization and performance.
Multiple works designing server consolidation solutions assume that the physical host is equipped with a single-core CPU. And multi-core processors have long been popular in personal entertainment, scientific research, and data centers. A CPU package consists of multiple dies, and each die encapsulates multiple cores. Due to the involvement of inter-core communication, inter-die communication, and other CPU components, the power consumption of a multi-core CPU is much higher than that calculated by the single-core CPU power consumption model. Therefore, the server consolidation model based on a single-core processor cannot accurately describe the user's energy demand. In addition, CSPs need to provide additional overhead to maintain VM migration in server consolidation and possible SLAV compensation. In this paper, we establish a server consolidation cost model based on the use of multi-core processor memory resources, VM migration, and SLAV compensation, and propose corresponding solutions to achieve a balance between cost and performance. Our contributions are as follows:
- We formally define a host power consumption model based on multi-core CPU and memory resource usage, and describe the cost of VM migration and SLAV on this basis. After proposing the cost model, we give the corresponding optimization problem.
- A Denoise AutoEncoder-based filter is used to denoise the VM workload trace. Subsequently, we use the SRU-based RNN method to predict the workload of VMs. Based on the predicted results, a host load detection strategy is proposed that considers both current and future load conditions.
- To minimize the total cost of server consolidation, we propose a VM selection strategy and a VM placement algorithm. These methods take into account the scheduling and placement of VMs between different cores of the same CPU and between different CPUs of different hosts, as well as the current and future requirements of VMs for different resources.
- We conduct simulations to evaluate the performance of our proposed solution MMCC. The simulations’ results indicate that MMCC can reduce host energy consumption by 10% ~ 43.9\%, SLAV cost by 33.5% ~ 51.7\%, and total cost by 20.9% ~ 34.4\% as compared to the baseline methods.”
Point 4:- Figures 3-6, the included text is unreadable
Response 4: Thank you very much for your suggestion.
We have updated these figures’ positions and sizes in the revised manuscript. Please check Page 10 and 11. And these figures can be zoomed in to see more details.
Point 5:- Algorithm 2: lines (9,10,21,22) type mistake in "odered_cores"
Response 5: Thank you very much for your suggestion.
We have revised these mistakes in the manuscript, please check it on Page 17 (highlight parts).
Point 6:- Figure 12 is unclear and the included text is unreadable
Response 6: Thank you very much for your suggestion.
We have updated this figure’s position and size in the revised manuscript. Please check Page 22, and the figure can be zoomed in to see more details.
Point 7:- The conclusions section, line 409: what "reducing the CDC" means? this should be clarified
Response 7: Thank you very much for your suggestion.
We have corrected it to: “In this paper, we focus on reducing the total cost of server consolidation in CDC,…”. Please check Line 495 in the revised manuscript.

Reviewer 2 Report
This was a well-written article, and it describes a very practical contribution.
A couple of notes:
The acronym for Quality of Service (QoS), while very commonly used in this field, should still be defined. There should also be at least a short description of why this is an important consideration.
The abbreviation for Cloud Data Center was defined as CDC in the abstract but as CDD in the introduction.
Author Response
Response to Reviewer 2 Comments
Point 1:- The acronym for Quality of Service (QoS), while very commonly used in this field, should still be defined. There should also be at least a short description of why this is an important consideration.
Response 1: Thank you very much for your suggestion.
We have rewritten the introduction section from the second paragraph to describe the importance of QoS and highlight our contributions. Please check it below.
“Increasing the resource rate of cloud data centers (CDCs) is one of the most effective means to reduce management costs. But there is a conflict between reducing costs and the performance that cloud service customers get. To improve resource usage, virtual machines (VMs) or containers assigned to users must be highly concentrated on physical hosts. However, a high degree of centralization brings a high degree of resource competition. When the competition is too intense, the host may be overloaded, thereby reducing the performance and user experience of VMs. To ensure user experience, Service Level Agreement (SLA) is used to quantitatively describe the corresponding QoS. If the SLA cannot be maintained, the Quality of Service (QoS) is threatened and SLA violation (SLAV) is generated. When SLAV appears, cloud service providers (CSPs) need to provide compensation to users as punishment for failing to meet user performance requirements. Currently, server consolidation is used to dynamically adjust the load balance between hosts in a CDC. Server consolidation periodically checks the load of hosts in the cluster and initiates VM migration to achieve load balancing, thereby maintaining a balance between resource utilization and performance.
Multiple works designing server consolidation solutions assume that the physical host is equipped with a single-core CPU. And multi-core processors have long been popular in personal entertainment, scientific research, and data centers. A CPU package consists of multiple dies, and each die encapsulates multiple cores. Due to the involvement of inter-core communication, inter-die communication, and other CPU components, the power consumption of a multi-core CPU is much higher than that calculated by the single-core CPU power consumption model. Therefore, the server consolidation model based on a single-core processor cannot accurately describe the user's energy demand. In addition, CSPs need to provide additional overhead to maintain VM migration in server consolidation and possible SLAV compensation. In this paper, we establish a server consolidation cost model based on the use of multi-core processor memory resources, VM migration, and SLAV compensation, and propose corresponding solutions to achieve a balance between cost and performance. Our contributions are as follows:
- We formally define a host power consumption model based on multi-core CPU and memory resource usage, and describe the cost of VM migration and SLAV on this basis. After proposing the cost model, we give the corresponding optimization problem.
- A Denoise AutoEncoder-based filter is used to denoise the VM workload trace. Subsequently, we use the SRU-based RNN method to predict the workload of VMs. Based on the predicted results, a host load detection strategy is proposed that considers both current and future load conditions.
- To minimize the total cost of server consolidation, we propose a VM selection strategy and a VM placement algorithm. These methods take into account the scheduling and placement of VMs between different cores of the same CPU and between different CPUs of different hosts, as well as the current and future requirements of VMs for different resources.
- We conduct simulations to evaluate the performance of our proposed solution MMCC. The simulations’ results indicate that MMCC can reduce host energy consumption by 10% ~ 43.9\%, SLAV cost by 33.5% ~ 51.7\%, and total cost by 20.9% ~ 34.4\% as compared to the baseline methods.”
Point 2:- The abbreviation for Cloud Data Center was defined as CDC in the abstract but as CDD in the introduction..
Response 2: Thank you very much for your suggestion.
We have corrected this mistake in the abstract. Please check Line 2 in the revised manuscript.
